# Closing the gender gap at academic conferences: A tool for monitoring and assessing academic events

Carmen Corona-Sobrino[1,2]*, Mónica García-Melón[1], Rocio Poveda-Bautista[1], Hannia González-Urango[1]

1 INGENIO CSIC-UPV, Universitat Politècnica de València, Valencia, Spain, 2 Departamento de Sociologia I, Facultad de Ciencias Económicas y Empresariales, Universidad de Alicante, Alicante, Spain

☯ These authors contributed equally to this work.

* carcosob@ingenio.upv.es

**Data Availability Statement:** Data are available in the following public repository: Dataset - A tool for monitoring and assessing gender gap in

## Abstract

The importance of participation in academic conferences is well known for members of the scientific community. It is not only for the feedback and the improvement of the work, it is also about career development, building networks and increasing visibility. Nevertheless, women continue to be under-represented in these academic events and even more so in the most visible positions such as speaking roles. This paper presents the development of a tool based on performance indicators, which will allow monitoring and evaluating gender roles and inequalities in academic conferences in order to tackle the underrepresentation of women. The study identifies relevant perspectives (participation, organizational structure and attitudes) and designs specific lists of performance indicators for each of them. The tool is based on a combination of two multicriteria techniques, Analytic Hierarchy Process and Analytic Hierarchy Process Sort, and a qualitative analysis based on in-depth interviews and information gathered from a focus group. The use of the AHP multi-criteria decision technique has allowed us to weight the indicators according to the opinion of several experts, and with them to be able to generate from these weightings composite indicators for each of the three dimensions. The most relevant indicators were for the participation dimension. Additionally, the tool developed has been applied to an academic conference which has been monitored in real time. The results are shown as a traffic light visualization approach, where red means bad performance, yellow average performance and green good performance, helping us to present the results for each indicator. Finally, proposals for improvement actions addressed to the red indicators are explained. The work carried out highlights the need to broaden the study of gender equality in academic conferences, not only regarding the participation but also the performance of different roles and functions.

## Introduction: The gender gap in science

A lack of diversity limits the progression of science. Gender diversity is crucial, however, numbers reveal that gender inequalities persist [1, 2].

conferences (Version 1) [Data set]. Zenodo. http://doi.org/10.5281/zenodo.4277246.

**Funding:** Grant Number OR2019-60221 Funder: Open Society Foundations Programme: Open Society Initiative for Europe Award: Expanding the Female Talent Pipeline in Europe https://www.opensocietyfoundations.org/ The funders had no role in study design, data collection and analysis, decision to publish, or preparation of the manuscript.

**Competing interests:** The authors have declared that no competing interests exist.

The study of the gender gap in science has gradually broadened to include different perspectives. Some of them are well known, such as the different proportion of top-level researcher women working in R&D in Europe [2]. This is related to the so-called "leaky pipeline" effect, which refers to "where even subjects with gender parity (or even a majority of women) at undergraduate and postgraduate levels see declining proportions of women with increasing seniority, and very few in senior positions" [3].

In line with this perspective, a large number of existing studies have examined the persistence of gender imbalance in the scientific structure [1, 4–6]. It has been suggested that women have a lower wage [7, 8], are promoted more slowly [9], receive less research funding [10], and their proportion in editorial and faculties boards is lower [2]

A different line of studies of gender imbalance has focused on knowledge production. There is a wide choice of literature discussing the persistence of gender differences in the production and publication of scientific knowledge, such as bias in the journal reviewing process [11], differences in the number of citations of their work in comparison with their male colleagues [11], a gender asymmetry in collaborations [12] or even, a disparity in commenting on published academic research [13].

Although there are many studies considering gender inequalities in academia, research in the participation in academic conferences remains limited [3, 14–16]. Efforts have been made to highlight the critical under-representation of women [17, 18]. However, these previous studies can only be considered the first step towards a deeper understanding of the dimension of the gender gap in academic conferences.

Different research studies turn a spotlight on diverse aspects of women's participation in academic events but these studies have almost exclusively focused on one specific factor to analyse the gender differences in conferences, such as the proportion of female speakers [16, 19–21], the gender disparities in different leadership positions [22, 23] or the influence of female organizing committee members in the selection of the speakers [19, 24]. Also, previous studies concentrate on a single specific knowledge area, for instance, the proportion of invited women in the evolutionary biology discipline [14, 15, 20] or in the academic tourism discipline [23, 25], among others. One exception is Nittrouer's work which examines gender differences in colloquium speakers in six disciplines (biology, bioengineering, political science, history, psychology, sociology) [16]. To the best of the authors' knowledge, no previous research has either considered different dimensions that can occur in the analysis of female participation in conferences or can be applied to different knowledge areas. Therefore, our new approach is needed in order to study gender disparity in different areas of knowledge and to develop an in-depth study of different roles and types of participation.

This paper addresses the need for a combinative study of different dimensions of the gender gap in conferences, so far lacking in the scientific literature. We propose the development of a tool based on performance indicators, which will allow monitoring and evaluating gender roles and inequalities in academic events in order to tackle the underrepresentation of women. Since our main priority is to focus on mechanisms to enhance female representation in academic conferences, we propose indicators in a novel way using mixed methods with the support of the literature review and expert knowledge as a source of information and decision-making techniques.

Our goal is to identify all the relevant perspectives or dimensions related to the gender gap and to design specific lists of performance indicators for each of them. These measures will allow the organizers of academic conferences to monitor their performance according to each specific dimension. Performance indicators are supposed to shape behaviour and practices in some desirable direction, in our case into an academic events practice 'with no gender gap'.

For all the above reasons, gender gap should be treated as a multicriteria problem and therefore MCDA techniques are suitable for carrying out its monitorization. Thus, the

complexity of this tool based on performance indicators will be tackled with the combination of two multicriteria techniques: Analytic Hierarchy Process (AHP) and Analytic Hierarchy Process Sort (AHPSort). AHP allows us to assign weights to the indicators in order to develop composite indicators. Additionally, AHPSort classifies the conferences into the different levels of the criteria for their evaluation.

### Social roles theory

The absence of women in specific positions and developing specific roles might negatively influence the ambitions of other women [15]. It can generate a barrier in the sense of belonging to specific spheres, which sometimes are considered as "male fields".

Social roles theory provides a framework to understand the importance of being part of academic conferences. According to this theory, people try to figure out which characteristics are required to be successful in a given role by appraising the characteristics of the people who mainly develop that role [26]. The problem is that social roles underline gender stereotypes. These stereotypes "reflect perceivers' observations of what people do. If perceivers often observe a particular group of people engaging in a particular activity, they are likely to believe that the abilities and personality attributes required to carry out that activity are typical of that group of people" [27].

According to this theory, women (and maybe other under-represented minorities) may feel less capable, less prepared, or have less self-confidence to carry out certain roles [1]. Also, they may feel that they do not belong in that environment. Even more difficult could be the attribution of significant roles regarding such environments, for instance, participating as a keynote speaker in an academic conference. Consequently, if the gender balance in organising a conference is not taken into account, gender stereotyping in science will continue to be the norm [28].

This theory helps us with the idea of generating spaces which encourage and boost the visibility of women and it also helps future generations who can take them as a reference. In this line, Martin's [6] study underlines that "if we are going to encourage women into careers in science we need also to provide role models for them to aspire to. We need to show that being a woman and being a successful scientist are not mutually exclusive. One way of doing that is to give women scientists a platform to present their research".

For these reasons, the understanding of a broad representation of the different roles and behaviour for women in a conference needs to be addressed. The idea is to offer a tool that analyses what happens at conferences as regards to gender issues. Thus, such an understanding should contribute to the future of science where men and women can succeed regardless of gender [28].

### The importance of attending academic conferences

The importance of participation in academic conferences is well known among members of the scientific community. It is not only for the feedback and the improvement of the work but it is also about career development, building networks, and increasing visibility [3, 29]. Following Hinsley's work [3], conferences enhance the opportunities to increase a scientific reputation (perceive ability and scientific contribution) and also a social reputation (behaviour and professionals' relationships) in the scientific community.

Attending conferences is required for the successful development of an academic career. In line with the academic career model [30–32], an academic career is composed of three different interacting careers: cognitive, community, and organizational. The cognitive career consists of the production of scientific knowledge. It is composed of the publications and the

process to develop them (topics, publications practices, or projects, among others). The second one, the community career, is related to a series of stages and reputational status (apprentice, master, elite. . .) which are characterised by the increase of scientific autonomy, leadership, and responsibility. Finally, the organizational career consists of the sequence of positions which implies different salaries, access to infrastructures and resources, performance expectations and roles. These different careers develop simultaneously.

The academic career model [32] can be linked with the importance of attending and participating in academic conferences. Involvement in these events improves the quality of the work for the cognitive career, as well, enhancing self-esteem and individual visibility, and it implies an enrichment of the networks for the community career. Also, it is a prerequisite in some evaluation systems to promote to a different position in the organizational career. Thus, attending conferences is a crucial part of the development of a scientific career. Nevertheless, women continue to be under-represented in academic events [33, 34].

Gender balance in academic conferences should be a fundamental requirement. However, numbers reveal a difference in the proportion of female speakers [16, 19, 35], a disparity in presentation times by gender [15, 36, 37] or an imbalance in the number of female organizing committee members [19]. For instance, Schroeder [20] considers that the under-representation of women is partly attributable to a larger proportion of women, than men, declining invitations.

The explanation about the imbalance suggests that bias against women plays an important role in generating these differences [35, 38, 39]. There are conscious or unconscious biases, subtle or blatant, which have a great impact on the absence of female scientific models [1, 40]. The under-representation of women in science is a multi-dimensional problem. Strategies to face the gender gap should consider all these factors.

## The use of indicators to measure the gender gap

Some previous efforts have been carried out within the EU in order to define indicators on gender equality in the scientific realm, such as the monitoring Responsible Research an Innovation (RRI) policies [41, 42]. These approaches suggest that the focus should be on processes of institutional change to see whether general ambitions are translated into concrete forms of action. Furthermore, the European Commission publishes the *She figures* report [2], which is particularly addressed to policymakers, researchers and their employers, by monitoring human resources statistics and indicators in the research and technological development sector and gender equality in science. The GIM tool [43] is also based on indicators which seek to help partners assess complex gender dynamics, in the context of the implementation of the Agenda 2030.

While the indicator panels mentioned above provide a good overview of the participation of women and men in different sectors and at different levels, they do not seem to provide insight into the cultural issues associated with gender inequality in conferences. Similarly, they do not offer much insight into institutional arrangements and mechanisms for fostering gender balance in this specific goal.

In light of the diverse notions involved in the measurement of gender issues, women's participation becomes decisive. The difficulties related to gender policies include insufficient resources, resistance, lack of clarity [44] on the integration of different actors, and dependence on gender experts' knowledge in the development of a gender perspective in policy. Participatory dynamics have been suggested to overcome these risks and achieve the implementation of successful policies [45]. Therefore, the methodology chosen to define the performance indicators will have a participatory nature.

The final purpose of our new panel of performance indicators will be to support and provide evidence and arguments to inform political debates and policymaking in the field of gender equality in academic events.

The rest of the paper is organized as follows: In section 2 we present the research approach followed in the paper, in sections 3–7 the different steps of the proposed methodology are developed and explained, and finally in section 8 a reflection on the results and the conclusions of the research is carried out.

## Research approach

### Research ethics

The entire research process was approved by an official ethics committee. The data in this study were analysed anonymously. For the interviews and the focus group, we followed a two-stage process: anonymisation of whole transcriptions, and then anonymising individual data extracts. Participants signed an informed consent about study purpose, its aims and how the study data could be used.

For the AHP survey, all participants were informed about the purpose of the questionnaire and the confidentiality of their data. Participants were aware of their right to refuse to participate.

### Methodological approach

The proposed methodological approach of this research is presented in the following figure (Fig 1). It is developed through two main stages: the design of the general methodology and the application to a specific conference. The goal of the first one is to identify all the relevant perspectives or dimensions related to the gender gap and to design a specific list of performance indicators for each of them. The aim of the second one is to monitor the performance of a selected conference. All the indicators will be measured, and a traffic light visualization result will be obtained for each of them.

Finally, some recommendations and guidelines will be addressed to the organizers of academic conferences in order to improve their performance according to each specific dimension and to shape practices in some desirable direction into an academics events practice 'with no gender gap'.

In the following sections a detailed description of the procedure carried out is presented.

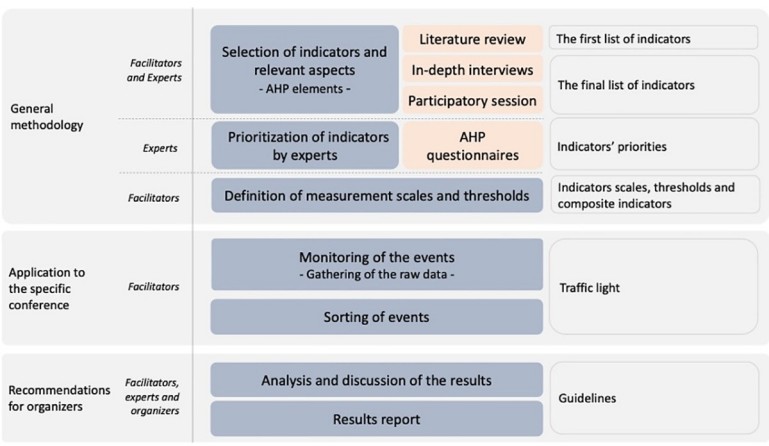

**Fig 1. Methodological approach.**

## Proposal of the first list of indicators based on a literature review

Different research studies (Table 1) turn a spotlight on diverse aspects of women's participation in academic events, such as the differences in exposure time between women and men, the importance of women's participation as a member of the organizing committee to provide greater opportunities for women to speak, or the idea of "token-women" to fill the female quota at events, among others. Our new approach is needed to study gender disparity in different areas of knowledge and to develop an in-depth study of different roles and types of participation.

The first list of indicators (Table 1) emerged after a comprehensive literature review on gender gap at academic conferences and events in order to identify which dimensions (or criteria) should be considered when monitoring the gender gap. We identified 4 different dimensions and 18 indicators, both qualitative and quantitative.

## Refinement of the list of indicators

The objective of this phase was to identify the more relevant and suitable indicators from the list obtained in the literature survey in order to produce a tailored reduced set of indicators. This phase was addressed in two steps with two different experts' groups: 1) In-depth interviews and 2) A participatory session.

### In-depth interviews with experts

We conducted seven in-depth interviews, online and real time, with gender experts and relevant academics to discuss the first list of indicators in order to identify the more relevant and feasible ones. Interviews attempted to cover a wide range of academic categories, different knowledge areas (social and technical visions) and institutional affiliations (Table 2). The main

**Table 1. First list of indicators arose from the literature review about gender gap in conferences.**

| DIMENSION | INDICATOR | REFERENCE |
|---|---|---|
| **FEMALE PARTICIPATION/ PRESENCE** | % of women who send abstracts and the ratio of acceptance | [35] |
| | % of women according to their job category/position and active participation | [1, 15] |
| | % of invited women to a baseline estimated using membership data of the associated scientific societies (or departmental members, academic staff, professional ranks. . .) | [16, 46, 47] |
| | % of women according to their institutional affiliation/nationality | [1] |
| | % of women invited as a keynote speaker according to bibliometric parameters (H Index or Impact Factor publications) | [33, 48], |
| | % of time exposition spent by a woman and the difference in the order | [15, 36], |
| | % of women who ask questions and the difference in the order | [3, 26] |
| | Gender differences in Twitter participation during the event | [49] |
| | % of women who receive prizes | [6, 47] |
| **LEADERSHIP** | % of women according to the type of active participation (Keynote/Plenary, Chair Moderator, Oral Speaker, Poster Speaker) | [23, 25, 50] |
| | % of women according to the distribution of organizers' roles | [23] |
| | % of women related to their contributions (poster, plenary speech. . .) | [35] |
| **ORGANIZATION CULTURE** | Female organizing committee members | [19, 24, 34] |
| | Event facilities regarding gender policies (family conciliation, breastfeed. . .) | [17, 19, 24] |
| | Ratio of participants who use family facilities | [6] |
| | Existence of an event conduct code | [6, 51] |
| **RESEARCH CONTENT** | % of track sessions and works which involve gender issues | [25, 52] |
| | Gender distribution of the authorship proposals | [53] |

**Table 2. Distribution of interviewees.**

|   | KNOWLEDGE AREA | CATEGORY | UNIVERSITY |
|---|---|---|---|
| 1 | Psychology | Full Professor | Utrecht University |
| 2 | Economy | Associate Professor | Toronto University |
| 3 | Sociology | Professor | Arizona State University |
| 4 | Economy | Associate Professor | Copenhagen University |
| 5 | Policy & History | Associate Professor | Utrecht University |
| 6 | Sociology | Full Professor | Barcelona University |
| 7 | Engineering | Associate Professor | Polytechnique University of Madrid |

selection criterion was academic relevance. In this sense, we understand academic relevance as a combination between the quality of research publications and projects and the attendance to academic conferences. In addition, interviewees were selected on the basis of opinions and recommendations from members of the corresponding scientific community regarding the most linkage with gender topics in their academic discipline.

Interviews were conducted between 2019 and 2020 and lasted around 40 and 60 minutes each, fully recorded and transcribed. They were analysed using qualitative content analysis [30] and then coded and managed using N-VIVO software for qualitative data analysis.

The semi-structured interviews focused mainly in the discussion of the dimensions and indicators. Additionally, interviewees offered very interesting information about each individual's trajectory and experience within the academic world, and on collective and individual strategies and ideas to tackle the gender gap in conferences. Table 3 shows a brief of the arguments presented by the interviewees according to the topic.

An illustration (Fig 2) is provided to exemplify the main ideas to reject and refine the list of indicators which were offered by our interviewees. Also, a new dimension was provided by them: "*Gender attitudes perception (gender behaviour, social dynamics, staging)*".

**Table 3. Main arguments exposed by the interviewees.**

| TOPICS | MAIN ARGUMENTS | QUOTES |
|---|---|---|
| Personal assessment for attending conferences | • It is a "must" in Academia<br>• Networking<br>• Obtain quality feedback<br>• Level and quality of the conference | "*You have to be presenting, and you have to make yourself known, and you need to participate. . . And all of those things are going to be harder*" (Professor) |
| Reasons behind the gender gap in conferences | • Prevalence of male networks<br>• Availability of women to participate (family & care obligations). Women decline invitations more often.<br>• Gender differences in funding resources<br>• Pipelines to form female talent<br>• The difficulty of finding female experts<br>• Conscious/unconscious bias for selection by males | "*We have had this problem where the organizers have invited all the keynotes, and then we say "hey, wait a minute, they are all white, they are all male, and they are all over the age of fifty, this is a problem*" (Associate professor) |
| Recommendations for organizers | • Avoid "all male panels"<br>• Expand the list of contacts and invite women first<br>• Re-think formats (one-day event, part/full register, virtual. . .)<br>• Conference scheduling<br>• Childcare facilities<br>• Take care of the small details | "*Inclusiveness is not only how many people are there. Counting them. Not only are they invited to the party, can they join, but also are they allowed to change the music? So, can they actually take part in the decision making, can they propose to do things in a different way?*" (Associate professor) |

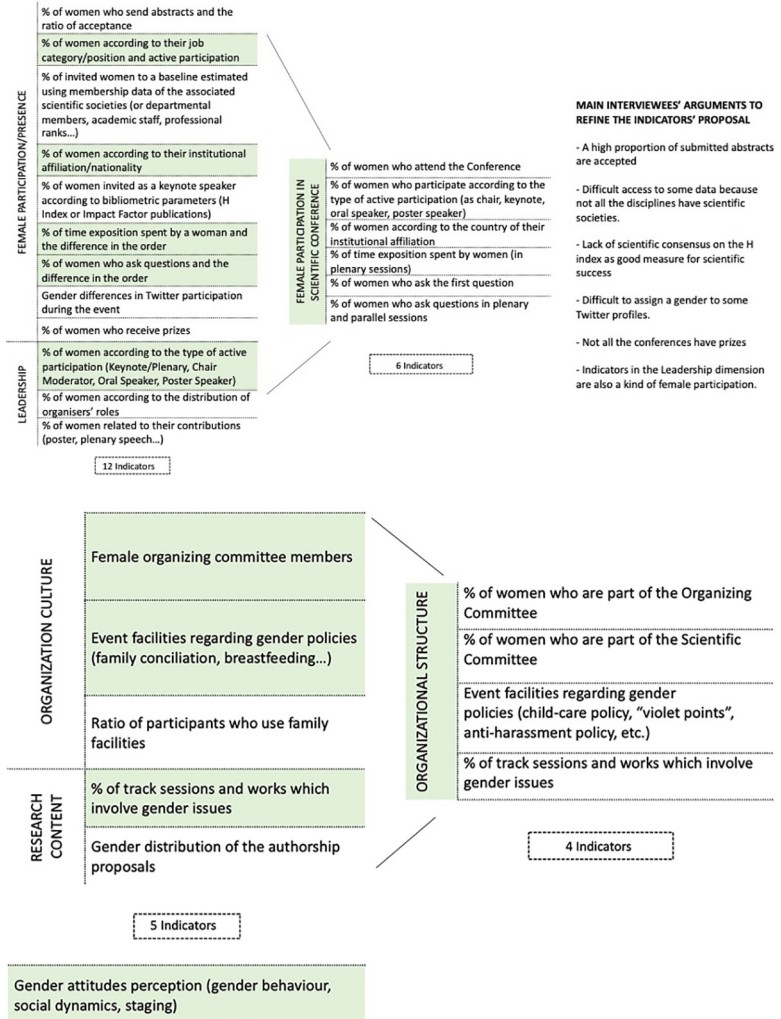

**Fig 2. Refining the list of indicators.**

### Participatory session and second list of indicators

A focus group discussion was organized in order to develop a deliberative process to integrate stakeholder values with technical judgments [54]. The discussion was conducted by two moderators with expertise in participatory decision-making methodologies and qualitative analysis.

For this focus group, a new set of experts has been selected, since in-person participation was required. Eleven participants took part in this focus group meeting (Table 4). They are all at different levels of their career path and have diverse disciplinary backgrounds, as well as different gender approaches (social and high-tech visions). Experts were selected on the basis of academic relevance (quality of their research publications and projects and the attendance to academic conferences). The main strength of the group was the merger in the different expertise, knowledge and visions with a solid methodological experience, which have provided an international and integrative vision.

The first list of indicators (Table 1) and the main ideas offered by our interviewees to refine the list of indicators (Fig 2) were presented to the participants. An in-depth discussion was oriented in order to analyse and reduce the list as well as the complexity of the tool. During this

**Table 4. Focus group participants.**

| | Knowledge area | Profile / Expertise |
|---|---|---|
| 1 | Economy | Research and innovation evaluation. Working for think tanks and government administration. Leading of multi-country European projects. |
| 2 | Political Sciences | Gender and Corporate Social Responsibility. |
| | | Researcher at a European Horizon 2020 project and Member of the gender commission of the Spanish observatory 'Mujeres, Ciencia e Innovación' (Women, Science and Innovation) for gender equality. |
| 3 | Chemistry | Public policies for scientific research and technological innovation, and the interactions between science and innovation. Extensive experience in management positions in various Spanish R&D institutions and leader of multiple national and international projects. |
| 4 | Psychology | Bibliometrics and cyber-metrics, databases, humanities information, information searching and retrieval, and scientific-technical information. |
| 5 | History | History of Science, history of medicine and cultural history. |
| | | One of the first women to lead an inter-university research institute in Spain. |
| 6 | Sociology | Gender studies in the area of women and equality policies. Feminist activist. |
| 7 | Engineering | Engineering and project management, evaluation of competitiveness, business strategy and evaluation of technology transfer activities of universities. |
| 8 | English language and literature | Contemporary world literature and education for sustainable development. |
| 9 | Economist | Evolutionary economics, economics of science and innovation science policy. |
| 10 | Engineering | decision-making processes, participatory decision-making and governance. Leading and working on projects on Responsible Research Innovation and Gender. |
| 11 | Management | Early-stage researcher working on feminist economy and grassroots social innovation. |

discussion, one new indicator appeared: *"% of women who attend the event"*. Dimensions and indicators were re-organized. Finally, the initial list was reduced to 11 indicators grouped in 3 dimensions. In Table 5. the resulting list is presented with a detailed description.

## Prioritization of indicators based on experts´ judgements. The use of AHP

The objective of this phase is to identify the more relevant indicators from the final list obtained in order to produce a tailored reduced set of indicators, built upon the hypothesis that there are gender indicators which are more important to be considered in certain academic conferences than in others.

Prioritization of gender indicators according to their level of importance has to be considered from an expert point of view [42]. Gathering and considering the different opinions and judgments of different gender experts can be a difficult task. While the literature deals extensively with the issues of gender indicators, it lacks a prescription of an easy-to-use, yet rigorous, methodology for ranking these indicators. Moreover, when the information available is biased and uncertain, as is the case of gender equality issues in academic events, it is necessary to make estimates. In such cases, experience and knowledge of the problem are more important than the prioritization model itself. Therefore, it is preferable to focus efforts on finding a renowned group of experts and get them involved in the process. For this purpose, we counted on the eleven participants of the focus group described in the previous step, who have diverse discipline and knowledge backgrounds and different gender approaches.

The results of the prioritization process will provide information about the importance of each indicator in order to reduce overall gender gaps in academic conferences. This will allow them to promote actions to improve their performance as regards these indicators.

**Table 5. Final list of indicators and definitions.**

| DIMENSIONS | INDICATORS | | DEFINITION |
|---|---|---|---|
| D1. Female participation in scientific conference/event | C1.1 | % of women who attend the conference | % of women who attend the academic conference according to the total of women who are part of the knowledge area (a standard classification) |
| | C1.2 | % of women who participate according to the type of active participation (as chair, keynote, oral speaker, poster speaker) | % of women who have an active participation, conducting sessions or making presentations, according to the total of women which are part of the knowledge area (a National standard classification) |
| | C1.3 | % of women according to the country of their institutional affiliation | % of women according to the total of assistants in each country of institutional affiliation |
| | C1.4 | % of time exposition spent by women (in plenary sessions) | % of women time exposition in plenary session according to the total of exposure time |
| | C1.5 | % of women who ask the first question | & & women who ask the first question according to the total number of first questions in all sessions. |
| | C1.6 | % of women who ask questions in plenary and parallel sessions | % of women who ask questions in plenary and parallel sessions according to the total of questions. |
| D2. Organizational structure | C2.1 | % of women who are part of the Organizing Committee | % of women who are part of the organizing committee according to the total of people who are part of the committee |
| | C2.2 | % of women who are part of the Scientific Committee | % of women who are part of the scientific committee according to the total of people who are part of the committee |
| | C2.3 | Event facilities regarding gender policies (child-care policy, "violet points", anti-harassment policy, etc.) | Qualitative indicator according to facilities provided by the event oriented to reduce gender gap |
| | C2.4 | % of track sessions and works which involve gender issues | % of sessions/tracks and works which involved gender issues according to all sessions. |
| D3. Gender attitudes perception (gender behaviour, social dynamics, staging) | | | Qualitative indicator based on attitudes and behaviours of gender perceived in the event |

The Analytic Hierarchy Process (AHP), a method proposed by Saaty [55], is based on theories of relative measurement of intangible criteria. AHP is a well-known technique that deconstructs a decision-making problem into several levels in such a way that they form a hierarchy with unidirectional relationships between levels. The top level of the hierarchy is the overall goal of the decision making problem. The following lower levels are the tangible and/or intangible criteria and sub-criteria that contribute to the goal. This technique allows us to deal with qualitative information about the criteria and its contribution to the objective of the decision making problem. The bottom level consists of the alternatives to be evaluated according to the criteria. AHP uses pairwise comparisons to allocate weights to the elements of each level, measuring their relative importance by using Saaty's 1-to-9 scale. The method also calculates a consistency ratio (CR) to verify the coherence of the judgements elicited, which should not be greater than 0.10 in order to be accepted. Mathematical foundations and steps of AHP can be found in Saaty [55, 56].

AHP has been chosen because it is one of the best known multicriteria analysis techniques and easy to understand by the experts who participate. There is little doubt that AHP has been frequently adopted. Along with its traditional applications, it has also been used lately in conjunction with others methods: data envelopment analysis (DEA) [57], fuzzy sets [58, 59], neural networks [60], SWOT-analysis [61] among others.

AHP has been used for the prioritisation of the final list of indicators shown in Table 5. The proposed hierarchy is very simple. At the top is the goal of the problem. In this case, our goal is "Prioritization of the gender criteria in order to tackle the under-representation of women in academic conferences." On the second and third levels are the three dimensions and the ten indicators for prioritizing, respectively. According to the AHP method, questionnaires were carried out to allocate weights by means of © 2007–2020 Expert Choice Software v.6.0.013.38933.

The AHP method was explained to the experts during the participatory session. Subsequently, an online questionnaire was sent to them. The answers to the questionnaire were collected by the ©Expert Choice Software. The results obtained were analysed with the help of this online software, which is widely used to support the resolution of AHP problems. Prioritization results for each individual expert were obtained according to their judgment. In order to obtain a group judgement, aggregation of individual judgments (AIJ) were performed using the geometric mean for all the experts [62, 63].

Weights of the criteria show the priority obtained for each indicator, a non-dimensional value that can be considered the relative importance of each one. Care was taken to ensure that all pairwise comparison matrices had a consistency ratio (CR) of less than 10%. It assesses the degree of inconsistency an expert has when eliciting their judgments. Whenever judgments were inconsistent, it was suggested that experts reconsider them so that they would fall within the acceptable limit.

### Weights of the dimensions. At the first level of the hierarchy

Table 6 shows the results obtained for each individual expert and the group results in this first level of the prioritization model. The group results have been obtained using the geometric mean of all the individual judgments in order to obtain the aggregation judgement, as it is recommended by Saaty [64]

The weights of the dimensions provide an important insight of each participant overall and an underlying conception of what gender equality means in academic conferences. Having the indicators classified in three dimensions with their weights makes it easier to adapt the evaluation tool to all possible conference contexts (face-to-face, on-line, hybrid, which is very interesting in this new pandemic era, when a significant increase in online conferences is expected.

The experts'priorities indicate that Female participation is more important than Organizational structure and that this last one is more important than Gender attitudes perception.

To analyse each particular conference the group weight will be used.

### Weights of the indicators. At the second level of the hierarchy

Weights of the sub-criteria at the second level of the hierarchy show the priority obtained for each indicator, a non-dimensional value that can be considered as the relative importance of each one. The global priority results obtained for all the group of experts were used in order to build composite indicators for the event (Table 7). With this global information of which indicators should be reinforced to improve the gender behaviour of the conference, guidelines can be provided.

The local priority results were used to explain how the tool works inside each dimension (Table 7).

The local priority results show the weights distribution of each indicator (at the second level of the hierarchy) with respect to the dimensions (at the first level of the hierarchy). With these results, we can analyse the predominant indicator within its dimension. As we can

**Table 6. Weights of the dimensions for each expert and aggregation of individual judgments.**

| Dimension | Expert 1 | Expert 2 | Expert 3 | Expert 4 | Expert 5 | Expert 6 | Expert 7 | Expert 8 | Expert 9 | Expert 10 | Expert 11 | Group (AIJ) |
|---|---|---|---|---|---|---|---|---|---|---|---|---|
| D1. Female Participation | 13,91 | 33,33 | 22,22 | 49,42 | 19,98 | 43,25 | 35,01 | 19,85 | 60 | 52,22 | 63,46 | 37,12 |
| D2. Organizational Structure | 43,48 | 33,33 | 65,02 | 25,48 | 42,22 | 14,46 | 40,15 | 45,46 | 20 | 14,57 | 25,43 | 33,52 |
| D3. Gender Attitudes Perception | 42,60 | 33,33 | 12,76 | 25,09 | 37,81 | 42,29 | 24,84 | 34,69 | 20 | 33,21 | 11,11 | 29,36 |

**Table 7. Weightings of the indicators.**

| Objectives | Local Priority | Global Priority |
|---|---|---|
| D1. Female participation in Scientific Conference | **37,12%** | **37,12%** |
| C1.1% of women who attend to the conference | 14,67% | 5,44% |
| C1.2% of women who participate according to the type of active participation | 34,95% | 12,97% |
| Keynotes | 27,18% | 10,1% |
| Moderators | 5,38% | 2,00% |
| Speakers | 2,39% | 0,90% |
| C1.3% of women according to the country of their institutional affiliation | 8,39% | 3,11% |
| C1.4% of time exposition spent by women (in plenary sessions) | 18,49% | 6,86% |
| C1.5% of women who ask the first question | 11,19% | 4,15% |
| C1.6% of women who ask questions in plenary and parallel sessions | 12,32% | 4,57% |
| D2. Organizational structure oriented to reduce Gender Gap in Scientific Events | **33,52%** | **33,52%** |
| C2.1% of women who are part of the Organizing Committee | 20,87% | 6,99% |
| C2.2% of women who are part of the Scientific Committee | 26,98% | 9,04% |
| C2.3 Event facilities regarding gender policies (child-care policy, etc.) | 33,79% | 11,33% |
| C2.4% of track sessions and works which involve gender issues | 18,36% | 6,15% |
| Keynotes | 15,78% | 5,30% |
| Tracks | 2,58% | 0,90% |
| D3. Gender attitudes perception (gender behaviour, social dynamics, staging) | **29,36%** | **29,36%** |
| **Goal** | **100,00%** | **100,00%** |

observe, the most important indicator for D1 Female participation in Scientific Conference is C1.2% of women who participate according to the type of active participation (34,95%) and specifically the case of keynotes speakers. Regarding the results for D2 Organizational structure oriented to reduce Gender Gap in Scientific Events indicators, C2.3 Event facilities regarding gender policies (child-care policy, etc.) is the predominant indicator (33,79%).

The global priority results show the weights distribution of each indicator with respect to the goal, i.e. with respect to 100% percentage. With these results, we can analyse the predominant indicator in general. Again, we can observe that C1.2 and C2.3 are the most important, both above 10%.

## Definition of measurement scales and thresholds for each indicator according to what each indicator assesses

After the relative importance of all the indicators has been obtained in the previous step of the methodology, we need to consider all aspects related to the feasibility of the actual application of these indicators. A detailed analysis of the capacity to collect data from the indicators will be conducted. Thus, an in-depth study to identify how best to collect the data required, and their corresponding measurement scales, is needed.

Measurement scales should allow us on the one hand to identify the measured value with a determined category or class (green, amber, red) and on the other hand to construct composite indicators for each dimension based on all the individual indicators. This in turn would allow the definition of labels for the three different outcomes of the composite indicators which will indicate the overall performance in gender gap of the conference.

To identify a value within a predefined category is a sorting problem. Values measured for the indicators should be sorted into the three different categories (good, average, bad) or according to our traffic light visualization method: green, amber and red.

## Measurement of indicators

To sort the values obtained in the monitoring process of the conference we propose to use AHPSort [65]. It is an extension of AHP [55] for sorting problems. It was proposed in 2012 by Ishizaka et al. [65] and has since been applied to several case studies [66–69]. AHPSort has the advantage of requiring far less pair-wise comparisons than the original AHP.

AHPSort is used for the sorting of alternatives into predefined ordered categories. The aim of this method is to classify the alternatives into the different categories of the criteria for their evaluation. The problem is structured using alternatives, criteria and categories for each criterion. The main steps of AHPSort can be summarized as follows:

1. Structuring the problem: to start with, the problem needs to be defined and structured as regards the classic AHP method: goal of the problem, criteria $cj$, $j = 1, . . ., m$, and alternatives $ak$ $k = 1, . . ., l$.

2. Establishing the classes: since it is a sorting problem, we need also to define the categories for each criterion that we called classes $Ci, i = 1, . . ., n$, where $n$ is the number of classes. The classes are ordered and have a label (good, average, bad). In order to properly define the classes, local limiting profiles $lpij$ have to be fixed, indicating the minimum performance needed for each criterion $j$ to belong to a class $Ci$.

3. Developing AHP pairwise comparisons: to continue, pairwise comparisons for each single alternative $ak$ with all the limiting profiles $lpij$ for each criterion $j$ are to be fulfilled. From these comparison matrices, the local priority $pkj$ for the alternative $ak$ and the local priorities $pij$ of the limiting profiles $lpij$ are calculated with the eigenvalue method [55].

4. The assignment to a class $Ci$ is done by comparing $pk$ with $lpi$. The alternative $ak$ is assigned to class $Ci$ which has the $lpi$ just below the global priority $pk$ [65].

$If\ p_k \geq lp_1 \rightarrow a_k \in C_1$

$If\ lp_2 \leq p_k < lp_1 \rightarrow a_k \in C_2$

$If\ p_k < lp_{n-1} \rightarrow a_k \in C_n$

For more details on this sorting method please see [65].

In our case the alternatives would be the conferences monitored, the criteria the list of indicators obtained and the categories of the three predefined classes for each indicator (green, amber and red).

For each of the classes a local limiting profile has to be fixed, indicating the minimum performance needed on each value measured for each indicator to belong to a class. Reference limiting profiles for the three categories are fixed according to the gender related literature and in equality studies published by the different European governments [2, 42, 70, 71]. They indicate the minimum level for a value measured to achieve the amber and the green classes that we called equilibrium threshold (amber) in the first place and the parity threshold (green) in the second place.

By means of AHP pairwise comparisons, each alternative obtains a local priority. To find out in which category the alternative falls, it will only be necessary to see what local priority it has obtained and to see where it is placed in relation to the local priorities of the limiting profiles: green, amber or red.

## Local limiting profiles

The local limiting profiles or thresholds are obtained after a literature review and consultation of gender policy documents [2, 41, 42, 72–75].

Two thresholds have been differentiated. The minimum performance to achieve the amber class is the *equilibrium* threshold (40%-60%) that is, there are at least 40 percent women. Meanwhile, the minimum to achieve the green class is the *parity* threshold (50%-50%), that is, there are at least 50 percent women. These values have to be considered always in relation to the total number of women who belong to a certain knowledge area. The classification in knowledge areas and the percentage of women in each of them have been obtained according to a political report [75].

## Conference monitored and results

The conference we have monitored is of the Innovation discipline and took place in Norway in January 2020. It was the pre-COVID19 period and therefore fully face-to-face. A total amount of 385 persons attended the conference: 147 women and 238 men.

The thresholds for each indicator were calculated according to two main sources:

- A database on information about parity figures in the different scientific areas. This conference is classified within the knowledge area of Social Sciences, whose parity threshold is 41,9% [75].

- The number of participants attending the conference and the type of participation.

### Results measured

In the following table (Table 8) we present the limiting profiles and measurements obtained for the different indicators.

### Results classified and their traffic light visualization

Each indicator has been assessed by applying the classification technique AHPSort (Fig 3). This technique allows us to classify the results obtained with the measurement of each indicator into one of the classes (red, amber or green). Following the AHP procedure, pairwise comparisons were fulfilled and the local priorities for each single one measured for the conference and for all the limiting profiles for each indicator were calculated. Comparisons were developed by organizers and participants of the conference. They are well aware of the evolution of the conference and the situation of its area of knowledge in terms of gender parity.

In relation to this conference and its results, we can conclude that the only dimension that gets the green light is D3 (Fig 3), which indicates that the attitudes observed in the "unofficial" dynamics of the conference were very positive and did not induce gender bias.

The D2 dimension, referring to the organizational structure, is the one that has the worst behaviour and gets a red light (Fig 3). This indicates that both the organizing committee and the scientific committee had a strong gender bias. All of their indicators except one have also obtained a red light, which means that individually they also have bad behaviour. In the next section, we will reflect on how to improve behaviour in this dimension.

Finally, D1 has obtained an amber light result (Fig 3). When looking at the individual results of its indicators we see that there are very different kinds of behaviour, which will have to be analysed individually in the next section for future recommendations. An amber result creates the expectation that with a little improvement of some of its weak points it could get a green result with some ease.

### Recommendations and future guidelines for the conference organizers according to the results obtained

We focus on giving some advice related to the indicators that have obtained the worst results. These recommendations have been gathered from the interviews, the literature search, the focus group, and the experience of the authors.

**Table 8. Limiting profiles by indicators of the monitored conference.**

| Indicators | | Green threshold | Amber threshold | Result measured | Source |
|---|---|---|---|---|---|
| C1.1% of women who attend the conference | | 164 women | 137 women | 147 women | Parity threshold (41.9%) [75] |
| C1.2% of women who participate according to the type of active participation (as chair, keynote, oral speaker, poster speaker) | Keynotes | 3 women | 2 women | 3 women | Parity threshold (41.9%) [75] |
| | Moderators | 36 women | 31 women | 27 women | |
| | Speakers | 131 women | 110 women | 94 women | |
| C1.3% of women according to the country of their institutional affiliation | Europe | 144 women | 121 women | 131 women | Parity threshold (41.9%) [75] |
| | N Europe | 64 women | 53 women | 61 women | |
| | S Europe | 29 women | 24 women | 32 women | |
| | E Europe | 9 women | 7 women | 7 women | |
| | W Europe | 43 women | 36 women | 31 women | |
| C1.4% of time exposition spent by women (in plenary sessions) | | 114 min | 91,2 min | 108 min | Regarding total exposure time and the number of keynotes (50–50) |
| C1.5% of women who ask the first question | | 35 first questions | 26 first questions | 44 first questions | Regarding number of women attending (37,6%) |
| C1.6% of women who ask questions in plenary and parallel sessions | | 76 questions | 101 questions | 95 questions by women | Regarding number of women attending (37,6%) |
| C2.1% of women who are part of the Organizing Committee | | 5 women | 3 women | 3 women | Parity threshold (41,9%) [75] |
| C2.2% of women who are part of the Scientific Committee | | 16 women | 14 women | 11 women | Parity threshold (50–50) [75] |
| C2.3. Event facilities regarding gender policies (child-care policy, "violet points", anti-harassment policy, etc.) | | Facilities or actions declared in writing or verified by observation in situ | Facilities or actions verbally declared during the conference. (Intentions) | Facilities or actions were not declared or observed during the conference. | Experts on gender. |
| C2.4. % of track sessions and works which involve gender issues | C2.4.1. Keynote presentation devoted to gender issues | At least 2 keynote presentations devoted to gender issues | At least 1 keynote presentation devoted to gender issues | 1 | Experts on the knowledge area of the conference. Conference participants in the last 5 years |
| | C2.4.2. Presentations on gender issues | 10% of the conferences | 5% of the conferences | 1/86 (1,2%) | |
| D3. Gender attitudes perception | | Not observed | Efforts carried out to avoid gender gap in social meetings | Extraordinary actions or gender inequality situations were not observed or perceived during social meetings | |

**C1.2% of women who participate according to the type of active participation (as chair, keynote, oral speaker, poster speaker).** To improve the number of moderators, the experts recommended that organisers of academic conferences should extend their contact networks beyond the usual circles. The need to achieve a balanced number (at least 40%) following the maxim "Gender equity is everybody's concern" [23] should be stressed.

Another important point to encourage a diverse participation is to rethink the registration format in the conferences [51]. According to the experts consulted, one of the main problems in attending and participating in events, apart from limited funds, is the extent of these. The interviewees raised the possibility of making daily registrations that would solve the problem of family conciliation and the scarcity of funds to attend conferences.

| | | | weight global | weight local | INNOVATION CONFERENCE | Green threshold | Amber threshold | Indiv. result | Comp. result |
|---|---|---|---|---|---|---|---|---|---|
| D1. Female participation in Scientific Conference | C1.1 % of women who attend the Conference | | 0,054 | 0,147 | 0,19 | 0,72 | 0,09 | | |
| | C1.2 % of women who participate according to the type of active participation (as chair, keynote, oral speaker, poster speaker) | Keynotes | 0,101 | 0,272 | 0,7 | 0,22 | 0,08 | | |
| | | Moderato | 0,020 | 0,054 | 0,07 | 0,65 | 0,28 | | |
| | | Speakers | 0,009 | 0,024 | 0,06 | 0,64 | 0,3 | | |
| | C1.3 % of women according to the country of their institutional affiliation | | 0,031 | 0,084 | 0,19 | 0,72 | 0,09 | | |
| | C1.4 % of time exposition spent by women (in plenary sessions) | | 0,069 | 0,185 | 0,31 | 0,62 | 0,08 | | |
| | C1.5 % of women who ask the first question | | 0,042 | 0,112 | 0,79 | 0,13 | 0,08 | | |
| | C1.6 % of women who ask questions in plenary and parallel sessions | | 0,046 | 0,123 | 0,27 | 0,65 | 0,07 | | |
| D2. Organizational structure oriented to reduce gender gap | C2.1 % of women who are part of the Organizing Committee | | 0,070 | 0,209 | 0,1 | 0,62 | 0,28 | | |
| | C2.2 % of women who are part of the Scientific Committee | | 0,090 | 0,270 | 0,07 | 0,65 | 0,28 | | |
| | C2.3 Event facilities regarding gender policies (child-care...) | | 0,113 | 0,338 | 0,08 | 0,66 | 0,25 | | |
| | C2.4 % of track sessions and works which involve gender issues | keynotes | 0,053 | 0,158 | 0,2 | 0,6 | 0,2 | | |
| | | track | 0,009 | 0,026 | 0,06 | 0,68 | 0,26 | | |
| D3. Gender attitudes perception | D3 | | 0,294 | | 0,67 | 0,21 | 0,12 | | |

**Fig 3. Classification of the results and their traffic light visualization.**

The improvement in the number of speakers was considered by the experts to be more complex according to certain areas of knowledge. The first important point they highlight is the importance of organizing diverse panels: "avoid all male panels" [76]. To avoid this problem, the following guidelines are given:

1. invite women in the first place, because they refuse more invitations to conferences [5, 20] So that if there is a cancellation, there is room for manoeuvre to continue inviting another woman,

2. send the invitations sufficiently in advance to favour the conciliation of family and work,

3. consult the databases of women experts and scientific communities,

4. expand networks and avoid eminently male networks, and

5. be aware that there are biases, both conscious and unconscious [4, 14, 77], in the selection of experts, it is important to identify them in order to avoid them.

**C2.1% of women who are part of the organizing committee and C2.2% of women who are part of the scientific committee.** For the people interviewed these indicators are important in order to achieve more diverse events. Experts propose that the substantial changes in favour of diversity in the organization of events, come from the presence of women in decision-making and participation bodies, such as these committees. As demonstrated by Casadevall et al. [34] there is a correlation between the presence of women in the organizing committee and their participation as speakers.

Similarly, to correct these indicators, experts recommend the need to expand professional networks to reach a greater number of female academics. In this case, they point out the need to include academics in the early stages of their academic career to avoid the effect known as "leaky pipeline" [17].

**C2.3 event facilities regarding gender policies.** This indicator seeks to facilitate issues of family conciliation but also considers those actions carried out in favour of gender equality. In this regard, the most common recommendation is to offer spaces for family conciliation (e.g. breastfeeding or childcare rooms).

The experts considered it necessary to have "anti-harassment" policies that are made explicit by the organizing committee, either online, in the program or in the development of the event itself, such as "violet points", so that all attendees are aware of their existence and, if necessary, can access them.

**C2.4% of track sessions and works which involve gender issues.** For the experts consulted, one of the ways to improve this indicator is to call for participation in the conference (Call for Abstract/Papers), in an inclusive manner, which encourages academics to participate and not to be excluded. Likewise, it is necessary to influence the participation of female academics in the early stages of their careers.

On the other hand, depending on the area of knowledge, it may be appropriate to include a specific "track" that addresses gender issues in the area of interest of the conference. It is also necessary to encourage the inclusion of a gender perspective in the whole research process [78].

## Conclusions

### General conclusions of the paper

This paper proposes a methodology for assessing the gender gap in academic conferences. The methodology consists of two parts, a general one, which fixes the indicators and their weights and can be used in any other conference, and a particular one in which these indicators have been measured for a specific conference in the field of social sciences and innovation.

With regard to the general methodology, we have been able to draw some conclusions. The use of multidisciplinary working groups has allowed us to have a more complete vision of the different approaches to the gender gap and thus to obtain a holistic and robust list of indicators.

We want to emphasize as being very relevant that these indicators have been classified in three dimensions: participation, organizational structure and attitudes, which allows their analysis both individually and combined between them. For example, in the case of online conferences, the third dimension could not be evaluated since it requires the presence of the evaluators at the event. Also, in the case of conferences where we are only allowed to attend but do not have access to the organizing team, only the first dimension could be evaluated. Therefore, having the indicators classified in three dimensions makes it easier to adapt the evaluation tool to all possible conference contexts, which is very interesting in the post-COVID19 era, when a significant increase in online conferences is expected.

In addition, the use of the AHP multi-criteria decision technique has allowed us to weight the indicators according to the opinion of several experts, again multidisciplinary, and with them to be able to generate from these weightings, composite indicators for each of the three dimensions. The most relevant indicators were for the participation dimension: the number of female keynote speakers (27.2%), for the organizational structure dimension: event facilities regarding gender policies (33.4%), and finally for the third dimension: gender attitudes perception, which would take all the weight since it is a unique qualitative indicator.

In this way, not only are results obtained with traffic light colours for the individual indicators, but also for each of the dimensions. Having each indicator and each dimension classified with a colour makes it much easier to see which indicators are performing well and above all which are not and need to be improved.

Regarding the particular application of the proposed methodology to a specific conference we can offer the following conclusions:

The assessment of each of the indicators has been carried out by applying the classification technique AHPSort. The use of this technique allows the classification into categories (traffic

light colours) of the results obtained with the measurement of each indicator. Its application requires the expert knowledge of the people who set the limiting profiles that will serve as thresholds for each category. Therefore, this last stage of the methodology is not above criticism when it comes to predicting its replicability. In the case of the conference worked on in this paper, the authors were able to obtain information from people in the organization and above all from people who were very familiar with the conference and had attended it in all its previous editions. This allowed them to make the pairwise comparisons required by the AHPSort technique without great difficulty. Therefore, in these conclusions we want to reflect this fact, which we think will not always be easy to achieve in future conferences that wish to be monitored.

In general, we also want to emphasize that the creation of a tool to monitor academic events requires being able to use it independently of the discipline. Therefore, the applicability of our tool favours its use in any discipline. Likewise, the tool also allows us to compare results between different conferences.

The way to achieve a balance in the conferences is given by intentional changes. This tool favours being able to concentrate on the weakest points of the conference and to carry out, intentionally, the required changes.

The work carried out highlights the need to deepen the study of gender equality in academic conferences, not only regarding the participation but also the performance of different roles and functions. Therefore, the development of a tool that takes into account the different dimensions, both of participation and representation, is fundamental.

## Specific conclusions of the case study

In general, we have drawn a series of conclusions from the results of this study:

- It is essential to make intentional changes in order to organize diverse academic events and conferences for the benefit of science and the academic community.

- It is necessary to generate spaces which encourage and boost the visibility of women and helps future generations to take them as a reference according to the social role's theory.

- It is necessary to encourage the participation of women at different levels, not only in attendance but also in participation, leadership and the development of different roles.

- It is important to make changes from within the organizational structures of the events, including more women in Organizing Committees and Scientific Committees. It would be convenient to carry out brief and specific training sessions or "guidelines" for the sessions' chairs in order to maintain a balance in the questions session, the control of times or other questions that can affect a balanced development of the event.

- It is essential to elaborate and make explicit the equality policies followed by the organization of the event, as well as "anti-harassment" policies and to make their action protocols visible.

## Future recommendations

- Avoid panels composed only of men or only of women. Work to achieve diversity understood in a broad way (gender, ethnicity, work categories, age. . .).

- Select and invite the female keynote speakers first, in order to avoid cancellations resulting in men-only panels.

- Take into account possible biases when making certain organizational decisions (distribution of spaces, order of intervention. . .).

- Be aware of the small details for the development of the event (the different heights of the platform, the differences in the tone of voice between women and men. . .).

- Rethink the format of the conferences so that they favour family reconciliation. Offer one-day registration possibilities [79].

## Limitations of the work

In the field work, a binary gender system (male-female) has been assumed in order to delimit and speed up research. However, the authors are aware of the different forms of expression and identification that can occur in this type of event. Likewise, other forms of ethnic diversity or social minorities have not been taken into account, due to the need to limit the research.

Owing to the need for physical presence in the sessions to record participation times, it has only been possible to measure this indicator in the sessions called keynotes.

The most complex part is the work with experts when considering all the pairwise comparisons required by the AHPSort. It will not always be easy to count on the collaboration of people linked to the organization of the event and with a historical vision of the evolution of the conference. That is why it is important to study the "terrain" beforehand and to have as many allies as possible within the conference organizing committee.

## Author Contributions

**Conceptualization:** Carmen Corona-Sobrino, Mónica García-Melón, Rocio Poveda-Bautista, Hannia González-Urango.

**Data curation:** Carmen Corona-Sobrino, Mónica García-Melón, Rocio Poveda-Bautista, Hannia González-Urango.

**Formal analysis:** Carmen Corona-Sobrino, Mónica García-Melón, Rocio Poveda-Bautista, Hannia González-Urango.

**Funding acquisition:** Mónica García-Melón.

**Investigation:** Carmen Corona-Sobrino, Mónica García-Melón, Rocio Poveda-Bautista, Hannia González-Urango.

**Methodology:** Carmen Corona-Sobrino, Mónica García-Melón, Rocio Poveda-Bautista, Hannia González-Urango.

**Project administration:** Carmen Corona-Sobrino, Mónica García-Melón.

**Software:** Rocio Poveda-Bautista.

**Writing – original draft:** Carmen Corona-Sobrino, Mónica García-Melón, Rocio Poveda-Bautista, Hannia González-Urango.

**Writing – review & editing:** Carmen Corona-Sobrino, Mónica García-Melón, Rocio Poveda-Bautista, Hannia González-Urango.

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
