## [Decision Letter · Decision Letter 0]

10 Nov 2020

PONE-D-20-31026

Closing the gender gap at academic conferences: a tool for monitoring and assessing academic events.

PLOS ONE

Dear Dr. Corona Sobrino,

Thank you for submitting your manuscript to PLOS ONE. After careful consideration, we feel that it has merit but does not fully meet PLOS ONE’s publication criteria as it currently stands. Therefore, we invite you to submit a revised version of the manuscript that addresses the points raised during the review process.

We look forward to receiving your revised manuscript.

Kind regards,

Dragan Pamucar

Academic Editor

PLOS ONE

Journal Requirements:

3. Please ensure that you refer to Figure 3 in your text as, if accepted, production will need this reference to link the reader to the figure.

4. We note you have included a table to which you do not refer in the text of your manuscript. Please ensure that you refer to Table 7 in your text; if accepted, production will need this reference to link the reader to the Table.

Reviewers' comments:

Reviewer's Responses to Questions

**Comments to the Author**

1. Is the manuscript technically sound, and do the data support the conclusions?

Reviewer #1: Yes

Reviewer #2: Yes

2. Has the statistical analysis been performed appropriately and rigorously? 

Reviewer #1: Yes

Reviewer #2: Yes

3. Have the authors made all data underlying the findings in their manuscript fully available?

Reviewer #1: Yes

Reviewer #2: No

4. Is the manuscript presented in an intelligible fashion and written in standard English?

Reviewer #1: Yes

Reviewer #2: Yes

5. Review Comments to the Author

Reviewer #1: The paper is good. Emphasize the scientific contribution in the introduction. Be careful when using the abbreviation for AHP Sort because 3 abbreviations AHP Sort, AHPSort, AHP-Sort are used in the work. Choose one and use it throughout the paper. Explain the results obtained by the AHP method.

Reviewer #2: In the first part, it is written that the interview was conducted with 8 experts. In the part with the AHP method, 11 experts are mentioned. Where does this difference come from? Are they the same people or has a new group been formed for the second round.

In Table 2. The numbers are not arranged in order

The list of 11 focus group participants states different disciplinary backgrounds. It is not clear on the basis of which criteria the participants were selected? Why there was a difference in relation to the group from the beginning of the research. It is necessary to supplement the work.

The difference in the number and types of disciplinary origin of the interview participants (initial examination) and the focus group is not explained.

Complement the work with a selection of works from 2020. year

Analyze the application of the classical AHP method and its modifications in the literature (fuzzy AHP, roughAHP, ...). For example. Application of fuzzyAHP operation DOI: 10.12700 / APH.17.3.2020.3.3

Explain how you calculated the value of "Gruop (AIJ)" in Table 6.

6. PLOS authors have the option to publish the peer review history of their article (what does this mean?). If published, this will include your full peer review and any attached files.

Reviewer #1: No

Reviewer #2: No

---

## [Author Response · Author response to Decision Letter 0]

20 Nov 2020

Dear Editor and Reviewers, 

We would like to thank you for your comments and recommendations, which we have tried to include in this new version of the paper. All the changes introduced by the reviewers’ suggestions have been highlighted in yellow along the text in the file “Revised Manuscript with Track Changes”

We will also try to answer all your comments one by one:

Journal requirements 

We have done it.

The repository information for our dataset is available in the next link: https://doi.org/10.5281/zenodo.4277246

3. Please ensure that you refer to Figure 3 in your text as, if accepted, production will need this reference to link the reader to the figure.

We have done it.

4. We note you have included a table to which you do not refer in the text of your manuscript. Please ensure that you refer to Table 7 in your text; if accepted, production will need this reference to link the reader to the Table.

We have done it.

Comments to the author

3. Have the authors made all data underlying the findings in their manuscript fully available?

The repository information for our dataset is available in the next link: https://doi.org/10.5281/zenodo.4277246

5. Review Comments to the Author

Reviewer #1. 

Be careful when using the abbreviation for AHP Sort because 3 abbreviations AHP Sort, AHPSort, AHP-Sort are used in the work. Choose one and use it throughout the paper.

Every different abbreviation was changed to AHPSort throughout the paper

Explain the results obtained by the AHP method.

The most relevant results obtained by the AHP method are explained so that the objective of the prioritization of indicators is better understood. We have added an explanation about these results in paragraph 397-409 (manuscript version) 

Reviewer #2. 

In the first part, it is written that the interview was conducted with 8 experts. In the part with the AHP method, 11 experts are mentioned. Where does this difference come from? Are they the same people or has a new group been formed for the second round?

There were two different groups of experts: one for the interviews and the other one for the focus group and the AHP technique. 

The first group was composed of 7 experts. In this case, interviews were conducted online and face-to-face. For the focus group and the AHP technique, a new set of experts has been selected, since in-person participation was required. In addition, a wider variety of academic profiles were included in this case. 

We have clarified this issue in the paper including a better description of the different participant groups in paragraph 253-258 and 283-291. 

In Table 2. The numbers are not arranged in order

We have corrected that and the numbers are now arranged in order.

The list of 11 focus group participants states different disciplinary backgrounds. It is not clear on the basis of which criteria the participants were selected? Why there was a difference in relation to the group from the beginning of the research. It is necessary to supplement the work.

We have added an explanation about the criteria for the selection for both interviews and focus group in paragraph 287-289 and 252-257. The main selection criterion was academic relevance. In this sense, we understand academic relevance as a combination between the quality of research publications and projects and the attendance to academic conferences. In addition, interviewees were selected on the basis of opinions and recommendations from members of the corresponding scientific community regarding the most linkage with gender topics in their academic discipline. For the focus group, the need to attend in person was a decisive factor. 

The difference in the number and types of disciplinary origin of the interview participants (initial examination) and the focus group is not explained.

The interviews sought to offer a first approach to the phenomenon studied. They focused mainly on the discussion of the dimensions and indicators related to gender issues. But also, on collective and individual strategies and ideas to tackle the gender gap in conferences according to each individual’s trajectory and experience. In this sense, a variety of profiles is needed but we consider more relevant the expertise related to the topic of gender. For this reason, interviewees were also selected on the basis of opinions and recommendations from members of the corresponding scientific community regarding the most linkage with gender topics in their academic discipline.

In the contrary, the objective in the focus group was to deepen and refine the definition of indicators. In this sense, it was necessary to cover all areas of knowledge with social and high-tech visions. The emphasis on the presence of the different areas of knowledge is fundamental for the correct and varied redefinition and prioritization of indicators. 

Complement the work with a selection of works from 2020. Year

We have reviewed different papers from 2020 and have also included them in our references. For example, a new study about the representation of women at otolaryngology conferences (Barinsky et al., 2020), a different approach to the gender gap in science taking in account the gender gap in commenting academic research (Wu, Fuller, Shi, & Wilkes, 2020) or an interesting paper (Tulloch, 2020) which analyses actions and policies of conferences in order to provide equality, among others. 

Analyze the application of the classical AHP method and its modifications in the literature (fuzzy AHP, roughAHP, ...). For example. Application of fuzzyAHP operation DOI: 10.12700 / APH.17.3.2020.3.3

A short review of the latest applications of traditional AHP and AHP in conjunction with other methods has been carried out and presented in paragraph 343-344. We have also included references of year 2020.

Explain how you calculated the value of "Group (AIJ)" in Table 6.

We have explained it right before Table 6.

Aggregation of individual judgments (AIJ) were performed using the geometric mean for all the experts (Saaty 2001; Saaty and Peniwati 2008).

When individuals act in concert and pool their judgments in such a way that the group becomes a new ‘individual’ and behaves like one, there is a synergistic aggregation of individual judgments. Furthermore, since the group becomes a new ‘individual’ and behaves like one, the reciprocity requirement for the judgments must be satisfied and the geometric mean rather than an arithmetic mean must be used. Treating the group as a new ‘individual’ with AIJ requires satisfaction of the reciprocity condition for the judgments. Aczel and Saaty (1983), have shown that when aggregating the judgments of n individuals where the reciprocal property is assumed, only the geometric mean satisfies the Pareto principle (unanimity condition) and the homogeneity condition (if all individuals judge a ratio n times as large as another ratio, then the synthesized judgment should also be n times as large). Thus, for AIJ, the geometric mean must be used. the geometric mean is more consistent with the meaning of both judgments and priorities in AHP.

In case the group structure is homogenous and decision makers are willing to act like one single individual, a synergistic AIJ is possible. Each decision maker conducts the pairwise comparisons by himself. Afterwards the geometric mean method (GMM) could be used to obtain the group judgment for each entry of the comparison matrices.

The AIJ procedure (Saaty 1989) is conducted by determining the mean of the individual judgments for each entry of the pairwise comparison matrices. If ari. j are the individual judgments of the group members DMr with r = 1,..., R by comparing element i with element j the aggregated pairwise comparison judgment A (i, j) is computed by the weighted geometric mean method (GMM): AGMM (i, j) = ∏Rr=1 (ari. j )

Thank you for receiving our manuscript and considering it for review. We appreciate your time and look forward to hearing from you. 

The authors.

---

## [Decision Letter · Decision Letter 1]

24 Nov 2020

Closing the gender gap at academic conferences: a tool for monitoring and assessing academic events.

PONE-D-20-31026R1

Dear Dr. Corona Sobrino,

We’re pleased to inform you that your manuscript has been judged scientifically suitable for publication and will be formally accepted for publication once it meets all outstanding technical requirements.

Kind regards,

Dragan Pamucar

Academic Editor

PLOS ONE

Additional Editor Comments (optional):

Reviewers' comments:

Reviewer's Responses to Questions

**Comments to the Author**

1. If the authors have adequately addressed your comments raised in a previous round of review and you feel that this manuscript is now acceptable for publication, you may indicate that here to bypass the “Comments to the Author” section, enter your conflict of interest statement in the “Confidential to Editor” section, and submit your "Accept" recommendation.

Reviewer #1: All comments have been addressed

Reviewer #2: All comments have been addressed

2. Is the manuscript technically sound, and do the data support the conclusions?

Reviewer #1: Yes

Reviewer #2: Yes

3. Has the statistical analysis been performed appropriately and rigorously? 

Reviewer #1: Yes

Reviewer #2: Yes

4. Have the authors made all data underlying the findings in their manuscript fully available?

Reviewer #1: Yes

Reviewer #2: Yes

5. Is the manuscript presented in an intelligible fashion and written in standard English?

Reviewer #1: Yes

Reviewer #2: Yes

6. Review Comments to the Author

Reviewer #1: All comments are accepted by the author. The paper and everything that was requested was corrected.

Reviewer #2: The mentioned suggestions were accepted and the necessary corrections were made in the paper.

Work can be accepted.

7. PLOS authors have the option to publish the peer review history of their article (what does this mean?). If published, this will include your full peer review and any attached files.

Reviewer #1: No

Reviewer #2: No

---

## [Editor Report · Acceptance letter]

26 Nov 2020

PONE-D-20-31026R1 

Closing the gender gap at academic conferences: a tool for monitoring and assessing academic events. 

Dear Dr. Corona-Sobrino:

I'm pleased to inform you that your manuscript has been deemed suitable for publication in PLOS ONE. Congratulations! Your manuscript is now with our production department. 

Kind regards, 

on behalf of

Dr. Dragan Pamucar 

Academic Editor

PLOS ONE